# The archerfish uses motor adaptation in shooting to correct for changing physical conditions

Svetlana Volotsky[1,2,3], Opher Donchin[1,2], Ronen Segev[1,2,3]*

[1]Department of Biomedical Engineering, Ben-Gurion University of the Negev, Be'er Sheva, Israel; [2]School of Brain Sciences and Cognition, Ben-Gurion University of the Negev, Be'er Sheva, Israel; [3]Department of Life Sciences, Ben-Gurion University of the Negev, Be'er Sheva, Israel

**Abstract** The archerfish is unique in its ability to hunt by shooting a jet of water from its mouth that hits insects situated above the water's surface. To aim accurately, the fish needs to overcome physical factors including changes in light refraction at the air-water interface. Nevertheless, archerfish can still hit the target with a high success rate under changing conditions. One possible explanation for this extraordinary ability is that it is learned by trial and error through a motor adaptation process. We tested this possibility by characterizing the ability of the archerfish to adapt to perturbations in the environment to make appropriate adjustments to its shots. We introduced a perturbing airflow above the water tank of the archerfish trained to shoot at a target. For each trial shot, we measured the error, i.e., the distance between the center of the target and the center of the water jet produced by the fish. Immediately after the airflow perturbation, there was an increase in shot error. Then, over the course of several trials, the error was reduced and eventually plateaued. After the removal of the perturbation, there was an aftereffect, where the error was in the opposite direction but washed out after several trials. These results indicate that archerfish can adapt to the airflow perturbation. Testing the fish with two opposite airflow directions indicated that adaptation took place within an egocentric frame of reference. These results thus suggest that the archerfish is capable of motor adaptation, as indicated by data showing that the fish produced motor commands that anticipated the perturbation.

*For correspondence:
ronensgv@bgu.ac.il

Competing interest: The authors declare that no competing interests exist.

## eLife assessment

This **valuable** study showed **convincing** evidence that archerfishes can adapt their shooting behaviors to airflow perturbations. The fish also exhibits adaptive behaviors indicative of an egocentric representation of the perturbation, though direct evidence is missing. Hence, this work will be of interest to those interested in cross-species comparisons for motor learning.

## Introduction

In a 1764 letter to the Royal Society of London, John Schlosser reported his discovery of a unique species (*Schlosser, 1764*). In this letter, Schlosser describes an amazing fish from Southeast Asia that can hunt insects above the water level by shooting a squirt of water from its mouth without ever missing. The archerfish, as it is called today, may not be as accurate as claimed (in fact, it misses quite often), but its behavior is nevertheless considered one of the most remarkable hunting strategies in nature. Of the many intriguing questions it raises, one remains conspicuously unresolved: how does the archerfish achieve accurate shooting despite various physical factors that affect the water jet?

**Figure 1.** The archerfish needs to correct for physical factors in shooting. The viewing angle of a target above the water level is shifted toward the zenith due to Snell's law. The fish needs to correct for this shift to hit the target in its actual position. The jet itself is affected by gravity and it does not proceed straight from the fish's mouth to the target. In addition, wind can affect the trajectory of the shot.

This question has been asked with regard to many types of motor behavior: How do animals make accurate saccades (*Shadmehr et al., 2010*; *Wallman and Fuchs, 1998*)? How do humans and other primates make accurate reaching movements (*Scheidt et al., 2005*; *Kluzik et al., 2008*)? In many of these cases, mechanisms of adaptation that respond to error and success manage aiming and calibration of the motor system (*Shadmehr et al., 2010*; *Donchin et al., 2012*). However, the question of maintenance of accurate motor behavior in fish has rarely been asked (*Tsvilling et al., 2012*; *Rossel et al., 2002*).

This study tests whether accurate shooting in the archerfish manifests characteristics that suggest underlying adaptation mechanisms similar to those seen in other systems. That is, first, can we show gradual change in the shot direction that reduces error in response to a perturbation? Second, will there be an aftereffect after the perturbation is removed?

During a shot, the fish's mouth is in the air, while its eyes remain well below the water level. Thus, in order to hit the target, the fish needs to compensate for multiple physical factors that affect the shot. This includes light refraction at the air-water interface, wind, and angle (*Figure 1*). Even the altitude of the target affects the shot due to gravitation (*Rossel et al., 2002*; *Timmermans and Vossen, 2000*).

The mechanism behind this ability to compensate for the different physical factors is unclear (*Dill, 1977*; *Temple, 2007*). Possible hypotheses as to the mechanism the fish uses for this correction is based on positing that the compensation is pre-programmed and hardwired through innate models of the physics. While this explanation is conceptually simple, it is hard to believe that it is sufficient, since during growth of the fish, the physical characteristics of its motor and visual systems change continuously, which would require a prewired program to take an unlikely number of highly interacting variables into account.

An opposing hypothesis posits that the archerfish does not have any innate physical knowledge and instead has a neural mechanism that develops an internal model of the current conditions through experience (*Shadmehr et al., 2010*). In this case, accurate shooting would depend on ongoing adaptation to consistent errors in a manner similar to the adaptation mechanisms that have been extensively investigated in the study of other vertebrate motor systems (*Fernández-Ruiz and Díaz, 1999*; *Donchin et al., 2012*; *Martin et al., 1996*). In this way, through a series of trials and errors, the fish would correct for misses while trying to hit the target. One advantage of an adaptation mechanism is that it can overcome the parameter differences caused by the physical changes that occur throughout the fish's life cycle.

These two approaches may be complementary. Innate models rely on underlying neural circuits, and these neural circuits probably have mechanisms of adaptation and calibration. Thus, the question of how the archerfish corrects for refraction and other physical factors can in fact be reformulated as 'to what extent is this ability innate, and how is it adapted'?

Indeed, the process of compensation itself can reveal aspects of the fish's innate internal models. For instance, if compensation generalizes in an egocentric reference frame, this implies that the fish associates errors with factors connected to the body of the fish such as muscle fatigue or physical factors that do not change with reference frame such as refraction index. If the compensation generalizes allocentrically, this implies that the fish associates errors with factors connected to the environment reference frame such as wind.

One of the key concepts in investigations of the associations between vision and motor output is visuomotor adaptation (*Martin et al., 1996*; *Fernández-Ruiz and Díaz, 1999*), which is a broad term for any practice-related change or improvement in motor performance in response to a perturbation in a task based on visual input. Visuomotor adaptation returns behavior to baseline levels of performance usually within tens or a few hundred trials, depending on the particulars of the task. A classic example of visuomotor adaptation is prism adaptation (*Kitazawa et al., 1995*; *Martin et al., 1996*): a sensory-motor adaptation that occurs after the visual field has been artificially shifted laterally or vertically by a prism. During prism adaptation studies, individuals wear special prismatic goggles made of prism wedges. The individuals then engage in a perceptual motor task such as pointing to a visual target. After performing the task repeatedly, the individuals improve over time until their performance is comparable to performance before the prism goggles were worn. Visuomotor adaptation has a few defining characteristics that differentiate it from other mechanisms of error correction; namely, it is gradual, error-driven, and shows an 'aftereffect': a tendency for the corrected behavior to persist for some time after the perturbation has been removed (*Fernández-Ruiz and Díaz, 1999*).

Here, we explored visuomotor adaptation in the archerfish, based on the extensive literature studying visuomotor adaptation in humans (*Izawa and Shadmehr, 2011*; *Shadmehr and Mussa-Ivaldi, 1994*; *Krakauer and Mazzoni, 2011*; *Fernández-Ruiz and Díaz, 1999*). We examined whether the archerfish can adapt to perturbations in the environment which result in distortion to the shot by generating an airflow above the water level to mimic the perturbation to the properties of air-water refraction. We show that the archerfish is indeed capable of visuomotor adaptation, which is consistent with the hypothesis that this adaptation partially contributes to the ability of the fish to overcome the changing conditions that affects the shot.

## Results

To test the capability of the archerfish for motor adaptation, we examined its ability to adjust to a consistent perturbation generated by airflow above the water level (*Figure 2A and B*). First, we trained the fish to shoot at a food pellet placed on a metal net. Then, an airflow was directed such that it provided a backwind to the jet produced by the fish (*Figure 2C and D*) and thus generated an error in the shot (which we defined as a positive error). The direction of the error in the shot was measured manually from images taken with a high-resolution video camera (see Methods).

### Archerfish exhibit motor adaptation

In the first experiment the fish had to shoot at the target so that we could obtain a baseline for the accuracy of its shooting. Then, the airflow perturbation was turned on and the fish had to compensate using motor adaptation to the deflection of the jet (*Figure 3A and B*, see Methods). Finally, there was a washout period where the airflow was turned off. The error for each shot was measured.

*Figure 3C and D* presents the errors for seven different fish in representative sessions – three with the direction and four – against the direction of the airflow, and all sessions for two fish in *Figure 3E and F* – one in each direction. For further analysis, we used all sessions from the seven fish. We analyzed individual fish performance and the average performance of all fish (*Figure 4*). As expected, when the perturbation was not present, the errors were distributed around zero. This is indicated in the raw traces (*Figure 4A and B*) and in the average responses of the individual fish (*Figure 4C and D*) as well as the average response of all fish (*Figure 4C and D*).

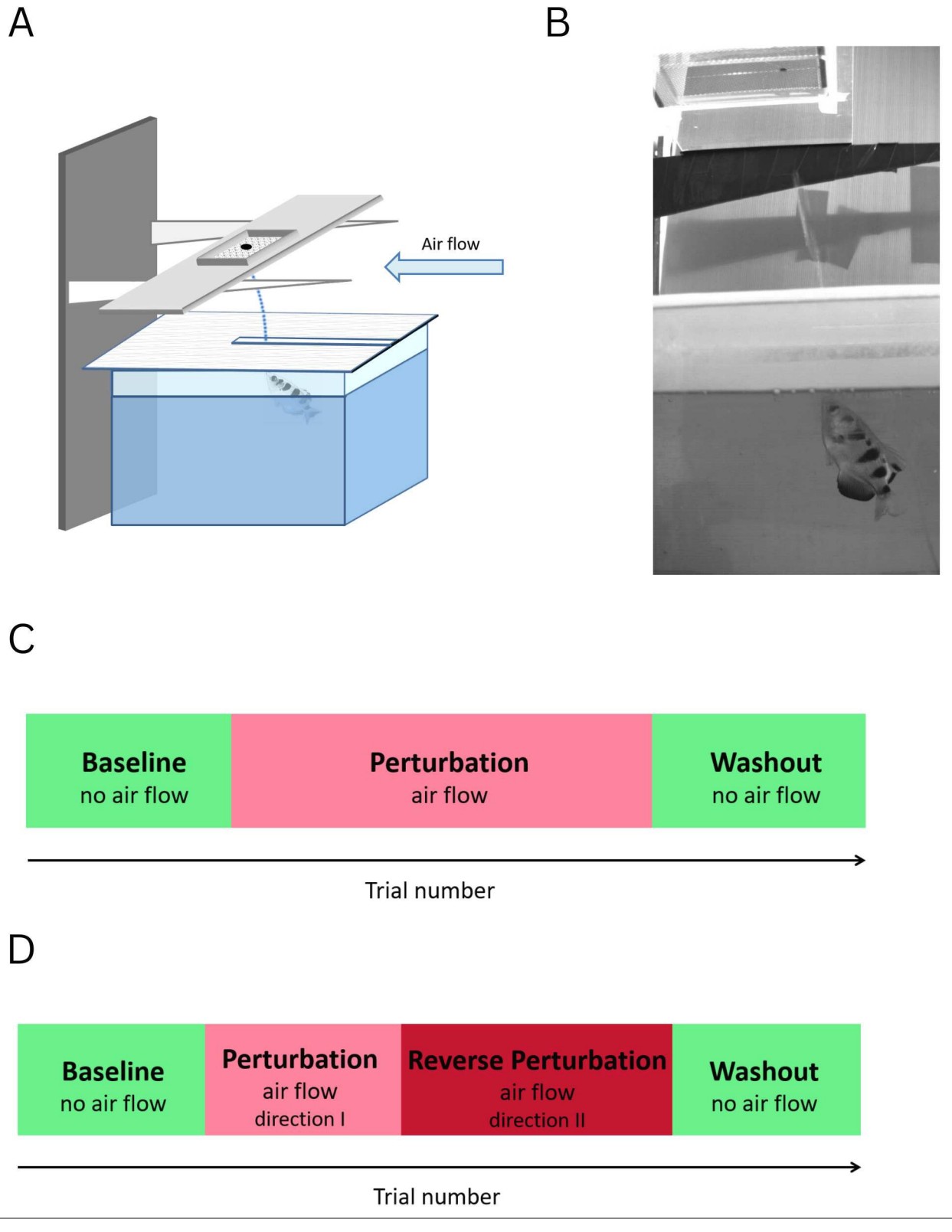

**Figure 2.** Schematic view of the experimental setup. (**A**) Water tank with a cover and a target above it. Airflow applied horizontally to the water's surface deflects the fish's shot. (**B**) An example from a video capturing the experiment. The airflow was oriented from right to left. The water jet is visible just before the impact at the target. (**C**) Experimental timeline of the first experiment: 5–10 shots before the introduction of the airflow, 10–15 shots with the airflow, 5–10 shots after the removal of the airflow. (**D**) Experimental timeline of the second experiment: 5–10 shots before the introduction of the airflow, 8–12 shots with the airflow in one direction, 15–20 shots with the airflow in the opposite direction, 5–10 shots after the removal of the airflow.

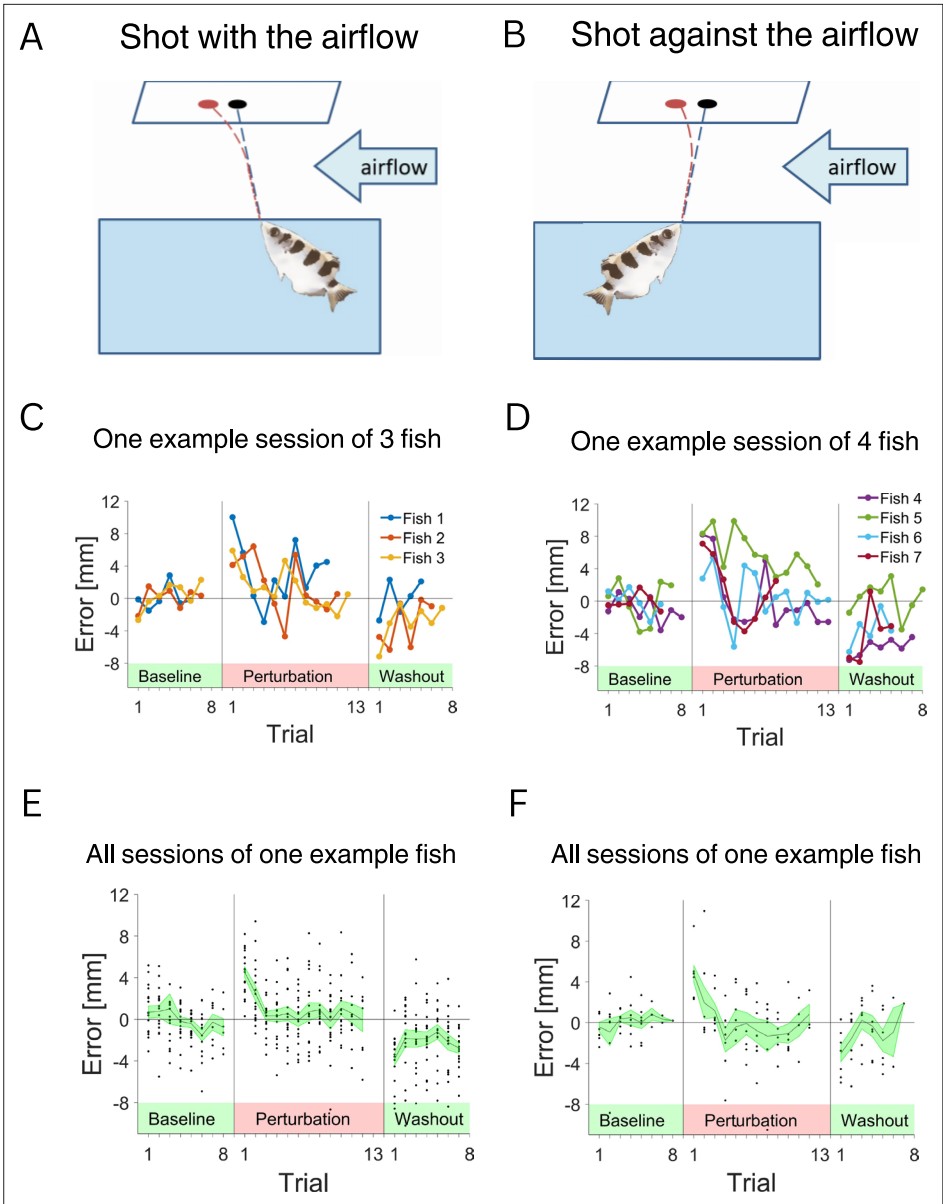

**Figure 3.** Examples of fish responses to the perturbation. (**A**) Experiment 1 setup: airflow in direction 1 – with the fish shot. (**B**) Experiment 1 setup: airflow in direction 2 – against the fish shot. (**C**) Example sessions for three fish that had to adapt to the perturbation in direction 1. Error was around zero during the baseline condition, increased with the introduction of the perturbation, and diminished with time. After the removal of the perturbation, the error was in the opposite direction. (**D**) Example sessions for the fish that had to adapt to the perturbation in direction 2. (**E**) All sessions for one example fish with the perturbation in direction 1. (**F**) All sessions of one example fish with the perturbation in direction 2.

After the airflow was turned on, the fish adapted to this perturbation. Immediately after the introduction of the airflow perturbation, there was a significant increase in shot errors (**Figure 4**, Perturbation, difference between epochs HDI (highest density interval) 4.28–5.52 mm, Cohen's d HDI 1.23–1.58). Then, over the course of several trials, the error became smaller and eventually plateaued. For all fish, the error at the beginning differed significantly from the error at the end of the perturbation period (difference between epochs HDI 3.1–4.33 mm, Cohen's d HDI 0.86–1.25).

After the airflow perturbation stopped (**Figure 4**, washout), the fish exhibited an aftereffect (**Figure 4**, Washout, difference between epochs HDI 2.65–3.99 mm, Cohen's d HDI 0.76–1.15). On the first trials after the airflow perturbation was removed, there was an error in the opposite direction

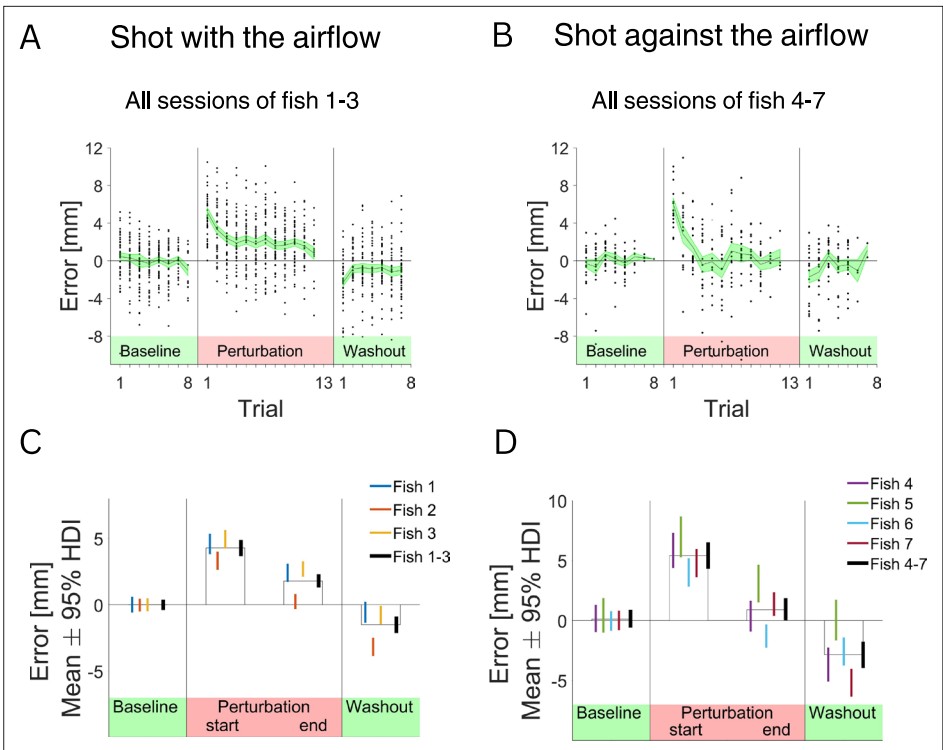

**Figure 4.** The archerfish can correct for perturbation using motor adaptation. (**A**) Response of three fish to the perturbation. Direction of the airflow is with the fish shots. Black dots are the error values across the sessions, the black line is the mean error value, and the SE is highlighted in green. (**B**) Response of three fish to the perturbation. Direction of the airflow is against the fish shots. (**C**) For three fish that shoot in direction 1: Mean value of the error and 95% highest density interval (HDI) for epoch 1 – the baseline, epoch 2 – first two trials after the introduction of the airflow, epoch 4 – last two trials before the termination of the perturbation, epoch 5 – first two trials after the termination of the perturbation. (**D**) For the four fish that shot in direction 2: Mean value of the error and 95% HDI.

that also decayed after several trials. This suggests that the fish generated motor commands that anticipated the perturbing airflow. The results were consistent for all seven fish in the experiment (*Figure 4*).

In our examination of the data, we sought out evidence of session-to-session 'saving' effect. However, the inherent noise within the collected data prevented us from drawing any definite conclusions.

## Motor adaptation is situated in fish egocentric frame of reference

Next, we assessed whether archerfish motor adaptation is based on an egocentric or allocentric frame of reference. The experiment was designed to let the fish adapt to airflow in one direction and then force the fish to switch direction by rotating the cover with the slot above the water tank, then shoot with the airflow in the opposite direction. There are no cues about the presence or direction of the airflow perturbation above the water tank; the activation of the air blower is done outside of the fish vision field. But the fish can see the visual landmarks inside the water tank, such as filter, plant décor, and thermostat. And they demonstrate a behavioral indication of position and direction within the tank in their shooting behavior (*Ben-Tov et al., 2018*; *Tsvilling et al., 2012*; *Dewenter et al., 2017*); when the cover with the slot above the water tank is rotated, the fish change their body position to shoot through the slot from the new angle.

This experiment was motivated by the fact that in our experimental setup, we introduced a perturbation, thus creating an environmental condition that remained constant while the change in direction relative to the perturbation was done by the fish. We hypothesized two outcome scenarios in response to the change. In the first, the fish would perceive the shift in allocentric reference frame: the

change occurring in its own position and the airflow perturbation remaining constant. The fish could use this information to correct for the true direction of the airflow and produce a smaller error.

In the second scenario, the fish would make a motor adaptation based on an egocentric reference frame: the environment changes together with the body position. In this case, the correction after the switch would present a large error since the correction for the previous perturbation would be added to the effect of the airflow. Motor adaptation in an egocentric reference frame is consistent with using the process to overcome refraction or other body-linked physical factors since they will be identical in both directions. Other physical factors, such as wind, would not be compensated correctly in an egocentric reference frame.

We put this hypothesis to the test and forced the fish to reverse the direction of their first shot while the direction of an airflow remained constant (*Figure 5A*). As in the first experiment, initially the response to the perturbation was a decrease in the error such that the fish hit the target toward the end of the trials (*Figure 5B–E*, Perturbation, difference between epochs HDI 2.47–5.05 mm, Cohen's d HDI 0.45–0.71). Then, immediately after the change in direction, the magnitude of the error increased considerably (difference between epochs HDI 14.32–16.89 mm, Cohen's d HDI 2.02–2.38), and adaptation took more trials than the initial response (*Figure 5B–E*, Reverse perturbation). Thus, in the absence of clues about the airflow direction, the fish apparently did not perceive the airflow perturbation as constant. After the switch, the fish continued to correct for the perturbation in the original direction relative to their body position.

We reanalyzed the results of the experiments by building mixed-effects models, with fixed-effects factor for the epoch and random-effects grouping factor for the fish. In ANOVA test for the models, the effect of the epoch factor was found to be significant and large (Experiment 1: p<0.001, $\eta^2$=0.67; Experiment 2: p<0.001, $\eta^2$=0.31). We performed post hoc analysis of the models to compare errors in different epochs of the experiments. The significance was evaluated using t-test with Bonferroni adjustment to correct for multiple comparisons. Cohen's d was used for effect size evaluation. In both experiments, all transitions between the stages of the experiments affected the error (Experiment 1: epochs 1–2 – p=0.001, Cohen's $d$=1.46; epochs 2–4 – p=0.003, Cohen's $d$=1.04; epochs 4–5 – p=0.002, Cohen's $d$=1.74. Experiment 2: epochs 1–2 – p=0.005, Cohen's $d$=1.55; epochs 2–4 – p=0.036, Cohen's $d$=0.93; epochs 4–5 – p=0.0004, Cohen's $d$=4.12; epochs 5–7 – p=0.003, Cohen's $d$=2.21; epochs 7–8 – p=0.007, Cohen's $d$=1.95).

Thus, we concluded that the fish engaged in motor adaptation of the shot in an egocentric reference frame. The egocentric adaptation process indicates that the fish did not use information about the unchanged nature of the perturbation; in this case airflow. Rather, the fish motor adaptation was consistent with adaptation to the air-water interface, which is isotropic in nature; i.e., it does not change with the shooting direction of the fish.

## Discussion

We tested the existence of the motor adaptation process in archerfish as a possible explanation for its ability to overcome changes in the environment during shooting. We introduced a perturbing airflow above the water tank on a fish trained to shoot at a target. The airflow deflected the trajectory of the water jet and as a result the fish's water jet missed the target initially. Over the course of several trials, all the fish managed to modify their shooting direction and successfully hit the target, demonstrating their ability to adapt to changes in the environment. After the removal of the perturbation, we observed an aftereffect, where the fish made errors in the opposite direction. This result suggests the formation of internal model in the fish brain that anticipates the presence of the perturbation.

We also tested whether the distortion by the airflow perturbation was consistent with egocentric or allocentric adaptation. For instance, the distortion expected by light refraction at the interface would require egocentric adaptation where wind would require allocentric adaptation. Thus, in the second experiment, we kept the airflow constant while the fish was forced to rotate its direction of shooting. We reasoned that if the fish perceived the perturbation as constant, they would correct for the opposite direction after the position change, and we would see small errors and quick improvement. Alternatively, if the fish adapted to egocentric factors, it would not be affected by the direction of the view. Thus, there would be no need to make additional corrections after the initial adaptation. Hence, immediately after the directional switch, the fish should not change its shooting strategy and we should observe large error. As shown in *Figure 5B–E*, after the change in direction, there was a

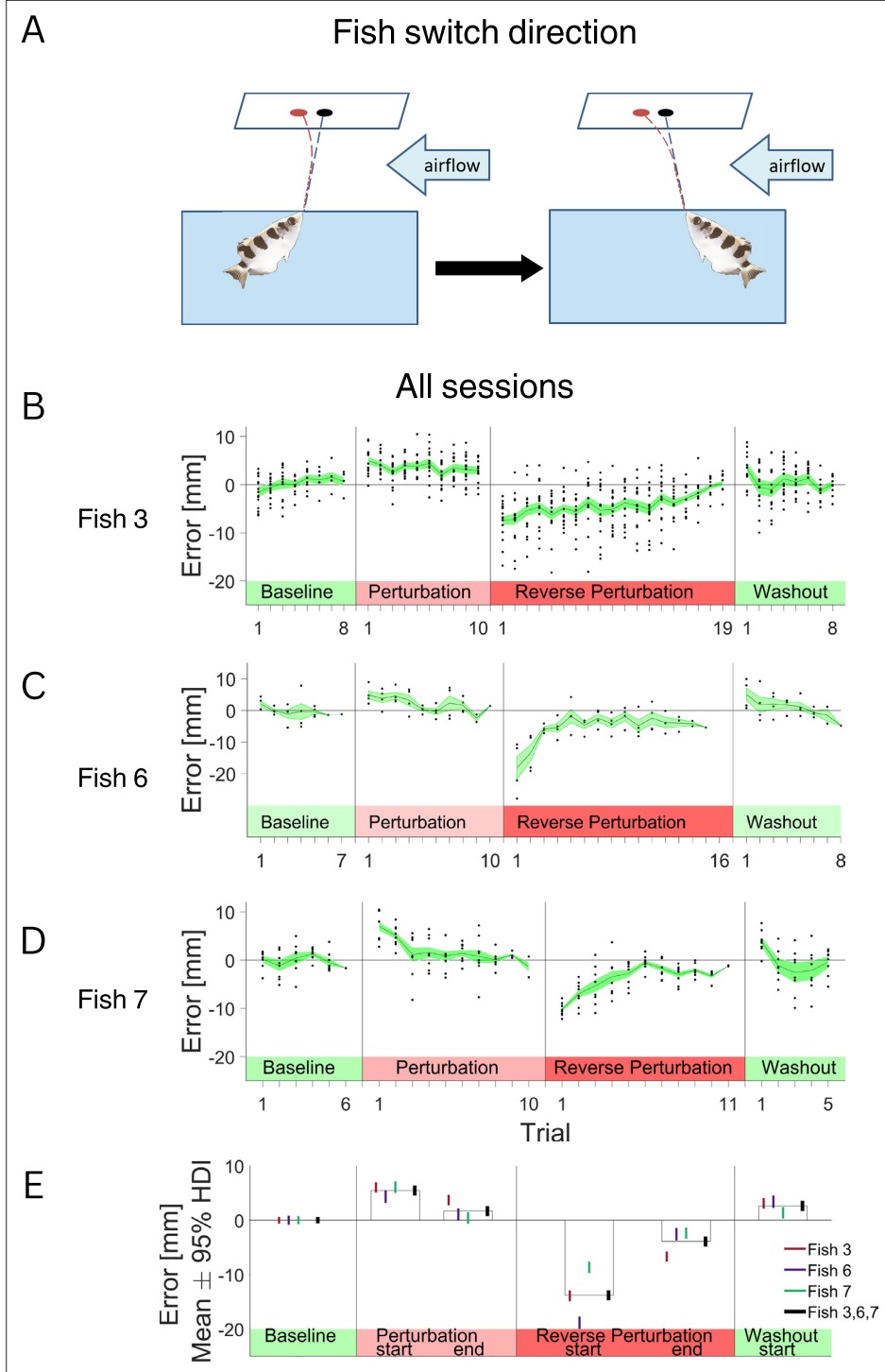

**Figure 5.** Adaptation in two directions sequentially reveals that motor adaptation is performed in the fish's egocentric reference frame. (**A**) Experimental setup: Airflow applied horizontally to the water's surface deflects the fish's shot. The fish first adapted to the perturbation in one direction and then enforced the switch direction of the shoot. (**B**) For the fish that changed direction from shooting in the direction with the perturbation to against the perturbation: mean error value and SE for the baseline trials, trials at the beginning and at the end of the adaptation period, trials at the beginning and the end of the adaptation period in the opposite direction and the beginning and the end of the washout period. (**C–D**) For the fish that changed direction from shooting in the direction against the perturbation to shooting with the perturbation: mean error value and SE for the baseline trials, trials at the beginning and the end of the adaptation period, trials at the beginning and the end of the

*Figure 5 continued on next page*

*Figure 5 continued*

adaptation period with reversed direction and at the beginning and the end of the washout period. (**E**) Mean value of the error and 95% highest density interval (HDI) for epoch 1 – the baseline, epoch 2 – first two trials after the introduction of the airflow, epoch 4 – last two trials before the change in the direction, epoch 5 – first two trials after the direction change, epoch 7 – last two trials before the termination of the perturbation, epoch 8 – first two trials after the termination of the perturbation.

pronounced increase in the error, and the decrease was slower than in the initial adaptation stage. This response to the change in the environment was consistent with the response to a change in the refraction index or other egocentric factors.

It is important to note that although it has been demonstrated that other fish species possess orientation and spatial memory (*Rodriguez et al., 1994*; *Cohen et al., 2023*; *Givon et al., 2022*; *Lee et al., 2012*; *Givon et al., 2023*; *Vinepinsky et al., 2020*), there is a lack of such information regarding archerfish specifically. The knowledge about other fish provides supportive evidence that archerfish may be sensitive to their relative positions in the water tank. However, it is essential to keep in mind that the current results do not completely rule out the possibility that archerfish are unaware of changes in their body position. Instead, they may continue with previously successful actions, which could appear as an egocentric generalization.

The two most important studies of archerfish ability to compensate for the distortion to vision at the air-water interface have focused on the phenomenology of compensation. Specifically, it was shown both experimentally (*Dill, 1977*; *Timmermans and Vossen, 2000*) and theoretically (*Barta and Horváth, 2003*) that the archerfish needs to compensate for the large distortion in the apparent elevation. Clearly, additional insights into how the archerfish aims at the target can be obtained by observing fish when they hatch and start shooting. To the best of our knowledge, there have been no reports of breeding archerfish in the lab or observation of the emergence of shooting behavior in the wild.

Our findings are consistent with previous work in the archerfish which evaluated the adaptability of the shoot behavior. These studies showed that the archerfish can predict the future position of targets (*Ben-Simon et al., 2012*), the future landing position of prey (*Rossel et al., 2002*; *Tsvilling et al., 2012*), and adjust their shot force during underwater behavior (*Dewenter et al., 2017*). Our findings extend these results to changes in the physical environment. Since most studies have used the archerfish visual system as a model for visual processing in general (*Newport et al., 2014*; *Newport et al., 2015*; *Ben-Simon et al., 2012*; *Mokeichev et al., 2010*; *Vasserman et al., 2010*; *Timmermans and Vossen, 2000*; *Temple et al., 2013*; *Volotsky et al., 2019*; *Volotsky et al., 2022*; *Segev et al., 2007*; *Ben-Tov et al., 2018*; *Ben-Tov et al., 2015*; *Gabay et al., 2013*; *Newport et al., 2018*), the current study contributes to a better understanding of a unique trait of the archerfish: the shooting mechanism and its control.

The need to correct for physical factors such as refraction at the interface is not unique to archerfish. For example, dwarf gouramis can shoot at targets above water and need to correct for refraction in the air-water interface (*Jones et al., 2021*; *Mann and Patterson, 2013*; *Miller and Jearld, 1983*). Other examples in the opposite direction of the water-air interface include works on the widespread hunting behavior implemented by birds to submerge prey by diving (*Machovsky Capuska et al., 2011*; *Machovsky-Capuska et al., 2012*).

This response profile in the archerfish is also consistent with the results of multiple studies of adaptation process in mammals (*Tseng et al., 2007*; *Shadmehr and Mussa-Ivaldi, 1994*; *Darmohray et al., 2019*; *Kitazawa et al., 1995*). When the sensory feedback to the executed movement differs from the expected feedback, animals can learn to adjust the strength and the direction of their subsequent movements until they are able to perform the task successfully. For instance, in studies of prism adaptation, subjects are asked to throw a ball at a target while wearing glasses with prisms that shift their visual field horizontally. For all subjects the initial error was shown to decrease after several trials; removal of the prisms caused disorientation and error in the opposite direction that also decreased with time (*Fernández-Ruiz and Díaz, 1999*; *Kitazawa et al., 1995*). Another important category of motor adaptation studies has investigated reaching movements when subjected to either visual distortions or opposing force fields (*Shadmehr et al., 2010*; *Kluzik et al., 2008*; *Shadmehr and Mussa-Ivaldi, 1994*). The results with respect to the error magnitude were similar and exhibited

a gradual decrease with time and the existence of an aftereffect. Finally, the study of visuomotor adaptation in mammals underscores the importance of the cerebellum as the neural site for this type of adaptation (*Darmohray et al., 2019*; *Donchin et al., 2012*; *Popa et al., 2016*). While the neuro-anatomy of the archerfish has been mapped (*Karoubi et al., 2016*), it remains unclear whether the archerfish homologue is similar.

Finally, it is important to note that not all fish subjects of the current study advanced to the final test, due to the challenging nature of the task and the difficulty in maintaining the fish's motivation throughout the test, which relies on feeding. Such challenges are common in behavioral studies. Consequently, this situation raises the question of whether all individuals are capable of solving the motor adaptation task. Addressing this question is a task for future studies.

Our work also suggests how the archerfish corrects for physical factors such as refraction through the use of motor adaptation as part of the correction process. In so doing, it sheds light on motor adaptation in vertebrates in general.

## Methods

### Animals

Wild-caught adult fish (6–14 cm in length; 10–18 g) were purchased from a local supplier. The fish were kept separately in 100 l aquaria filled with brackish water (salinity 6–8 ppt) at 25–29°C on a 12/12 hr light-dark cycle. Fish care and experimental procedures were approved by the Ben-Gurion University of the Negev Institutional Animal Care and Use Committee and were in accordance with government regulations of the State of Israel (protocol IL-47-08-2021(C)).

### Experimental scenery

The experiments were conducted in a water tank with a cover above the water that forced the fish to shoot through a slit in the cover such that the shot was aligned with the airflow perturbation when present (*Figure 2A and B*).

After delivery to the lab, the fish went through a period of acclimatization to the lab environment. Then, the fish were gradually trained to shoot at a black circle on a metal net placed 35–40 cm above the water, depending on the water level in the tank. After the fish were able to make 20–25 shots in 10 min, a plastic cover with a slot was placed above the water tank. The fish were trained to shoot at a target above the water tank to ensure uniform direction of the shot in relation to the direction of the later perturbation. The target was placed at the distance of 10 cm from the air nozzle. The speed of the airflow above the water was measured using an anemometer (SP-82AM, Lutron Electronics, Taiwan) just below the target. The air speed was 7.5 m/s±0.3 m/s.

### Training

Overall, 39 candidate fish were tested for the study. We first screened for the ability to persist shooting at least 20 shots per session through a slot in a cover above the water tank and remain accurate throughout the session. This was crucial since it enabled the collection of enough data for statistical analysis. Overall, seven fish passed the screening test and continued to the perturbation experiments.

Two of the seven fish were able to complete the 15 sessions. One fish completed 8 sessions, two fish – 7 sessions, one fish – 2 sessions, and one fish – 1 session in the first experiment. For the second experiment, the fish were required to perform at least 30 consecutive shots through a slot with a change in direction in the middle of the session. Only three fish met this criterion and were used in the experiment: one fish competed 15 sessions, one fish – 4 sessions, and one fish – 8 sessions.

### Experiment 1

The experiment consisted of three stages (*Figure 2C*). First, the fish performed 5–10 trials with no airflow. The number of trials under baseline conditions was generated from a uniform distribution so that the fish would not anticipate the change in conditions after a fixed number of trials. Then, we introduced a perturbing airflow horizontally to the water's surface that deflected the fish's shot trajectory and caused the fish to miss the target. The airflow was either in the direction of the shot or opposite to the direction of the shot. The fish performed 10–15 trials. Then the airflow was removed and the fish shot again under the baseline condition. Thus, the errors in the first and final stage

permitted comparisons of fish behavior in normal and post-adaptation conditions in the same physical environment.

## Experiment 2

We tested whether the fish adapt to the airflow perturbation within an egocentric or allocentric frame of reference. In the egocentric frame of reference, any perturbation is perceived as constant regardless of the subject position; e.g., refraction index on the water surface. In the allocentric frame of reference, the perturbation perception changes with the change in position; an example for this scenario is a wind above the water surface that affects the water shot differently depending on the fish position relative to the wind direction. The second experiment consisted of four stages: first, the fish performed 5–10 trials without the airflow to estimate the baseline; then, 10 trials with the airflow in one direction were administered (*Figure 2D*). Next, the fish were forced to reverse the direction of their shot by rotating the cover with the slot by 180 degrees with no change in the airflow direction. The airflow was turned off after 15–20 trials and the fish performed five additional shots with no perturbation. Throughout the experiments, the fish were given no clues about the initiation or termination of the airflow, nor about its direction.

## Video recording

The experiments were recorded using three high-resolution cameras (ISG LightWise Allegro, Imaging Solutions, USA) at a frame rate of 190 frames/s. One camera was focused on the target and its immediate surrounding area. Two other cameras recorded the fish from different angles to capture their behavior. The video clips were analyzed offline.

## Measuring shot error

We characterized the adaptation process in terms of the shooting error measured on each trial shot. The error was defined as the distance between the center of a target and the center of the water jet produced by the fish. The data were extracted from the movie frames of the camera that was focused on the target and its immediate surroundings.

## Statistical analysis

We used a Bayesian approach to analyze the adaptation process. To analyze the results of the first experiment, we defined six epochs in the timeline of the experiment: 1 – baseline trials; 2 – first two trials after the initiation of the airflow perturbation; 3 – trial from the third up to the two trials preceding the end of the perturbation; 4 – the last two trials before the termination of the perturbation; 5 – the first two trials after the termination of the perturbation; 6 – the remainder of the trials until the end of the session.

To analyze the results of the second experiment we defined nine epochs in the timeline of the experiment: 1 – baseline trials; 2 – first two trials after the initiation of the airflow perturbation; 3 – trial from the third up to the two trials before the direction changed; 4 – the last two trials before the direction changed; 5 – first two trials after the direction changed; 6 – trial from the third up to the two trials preceding termination of the perturbation; 7 – last two trials before the termination of the perturbation; 8 – first two trials after the termination of the perturbation; 9 – the remainder of the trials until the end of the session.

We performed a hierarchical Bayesian analysis to evaluate the behavior of the fish in response to the perturbation. The statistical analysis was conducted using R 4.0.4 programming language (https://www.r-project.org/) and the JAGS 4.3.0 statistical package (https://mcmc-jags.sourceforge.io/) (*Plummer, 2003*). JAGS was used to produce samples from the posterior probability distribution based on the data for the parameters of the statistical model described below (*Kruschke, 2014*).

The central parameter of interest was the shooting error of each fish in each time epoch. In our model, this error was drawn from a normal distribution. The mean of this distribution was modeled hierarchically as a linear combination of parameters that depended on the trial epoch, the subject, and the interaction between them. That is, the model of the error was written in the following way:

$$error_{trial} \sim Normal\left(a_{baseline} + a_{epoch} + a_{subject} + a_{epoch\ \&\ subject}, \sigma^2\right) \qquad (1)$$

where $a_{baseline}$ is the average of all baseline errors, $a_{epoch}$ is the average contribution to the error during a specific epoch for all fish, $a_{subject}$ is the fish's unique characteristics, and $a_{epoch\ \&\ subject}$ is contribution to the error due to the interaction between subject and epoch. Finally, $\sigma^2$ is the global variance. The model was hierarchical in the sense that the prior distributions of these parameters were themselves modeled using parametric distributions as follows:

$$a_{baseline} \sim N\left(\mu_{all\ trials}, \sigma^2_{all\ trials}\right)$$

$$a_{epoch} \sim N\left(\mu_{epoch}, \sigma^2_{epoch}\right)$$

$$a_{subject} \sim N\left(\mu_{subject}, \sigma^2_{subject}\right)$$

$$a_{epoch\&subject} \sim N\left(\mu_{epoch\&subject}, \sigma^2_{epoch\&subject}\right) \tag{2}$$

The posterior distributions of the hyper-parameters ($\mu$, $\sigma^2$) were determined simultaneously and jointly with the posterior distribution of the parameters in the linear model. The priors of the hyper-means were broad normal distributions and the priors of the hyper-variances were broad gamma distributions. The prior for the global variance parameter in the linear model was a broad uniform distribution.

Bayesian data analysis combines the data and the model to generate samples of the posterior distribution of the parameters given the data. JAGS carries out this sampling using a version of the Markov Chain Monte Carlo (MCMC) algorithm for sampling arbitrary target distributions. We generated three chains of 10,000 MCMC samples from the joint posterior probability distribution of all the parameters. The standard procedure is to use three or four chains to show that they converge to a similar result to verify the robustness of the outcome. Convergence of the algorithm and sampling properties were tested using both graphical and quantitative methods (*Kruschke, 2014*).

## Testing for significance

We used the 10,000 MCMC samples to calculate the 95% HDI. HDI is the range of values for which there is a 95% posterior probability of finding the parameter and where all the values within the interval are more probable than any value outside of it. By examining the 95% HDI of each fish in each time epoch, we could describe the different stages of the adaptation process in the archerfish. If the 95% HDI of the difference in the error between two epochs was greater than the region of practical equivalence of 5% around zero, the error was considered to be significantly different. For every comparison we report the HDI of the difference between the epochs in mm.

## Determination of effect size

Effect size was calculated for the difference between transition stages of the experiments: for experiment 1 – for the differences in error between epochs 1 and 2, 2 and 4, 4 and 5. For the second experiment, we compared error between epochs 1 and 2, 2 and 4, 4 and 5, 5 and 7, 7 and 8. For every MCMC sample, Cohen's d effect size was calculated as the difference between the mean values of the two compared instances divided by the pooled standard deviation. Then, 95% HDI was calculated for the values of effect size. Effect sizes are reported for all comparisons as the HDI of Cohen's d.

## Acknowledgements

We thank Mor Ben-Tov and Ehud Vinepinksy for help in the initial stages of this project. We gratefully acknowledge financial support from THE ISRAEL SCIENCE FOUNDATION – FIRST Program (Grant no. 281/15), THE ISRAEL SCIENCE FOUNDATION (Grant no. 824/21), and The Human Frontiers Science Foundation Grant RGP0016/2019.

# Additional information

### Funding

| Funder | Grant reference number | Author |
|---|---|---|
| Israel Science Foundation | First 281/15 | Opher Donchin Ronen Segev |
| Israel Science Foundation | 824/21 | Ronen Segev |
| Human Frontier Science Program | RGP0016/2019 | Ronen Segev |

The funders had no role in study design, data collection and interpretation, or the decision to submit the work for publication.

### Author contributions

Svetlana Volotsky, Formal analysis, Investigation, Visualization, Methodology, Writing - original draft, Writing - review and editing; Opher Donchin, Conceptualization, Supervision, Funding acquisition, Investigation, Methodology, Writing - review and editing; Ronen Segev, Conceptualization, Formal analysis, Supervision, Funding acquisition, Investigation, Methodology, Writing - original draft, Writing - review and editing

### Author ORCIDs

Svetlana Volotsky (iD) http://orcid.org/0000-0002-3086-573X
Ronen Segev (iD) http://orcid.org/0000-0002-8109-1076

### Ethics

Fish care and experimental procedures were approved by the Ben-Gurion University of the Negev Institutional Animal Care and Use Committee and were in accordance with government regulations of the State of Israel (protocol IL-47-08-2021(C)).

Reviewer #1 (Public Review): https://doi.org/10.7554/eLife.92909.3.sa1
Reviewer #2 (Public Review): https://doi.org/10.7554/eLife.92909.3.sa2
Author response https://doi.org/10.7554/eLife.92909.3.sa3

---

# Additional files

### Supplementary files

• MDAR checklist

### Data availability

All data and code supporting the findings of this study are openly available at (https://github.com/volotsky-s/motor-adaptation-archerfish, copy archived at *Volotsky, 2024*). This includes raw data and the code used for analysis.

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
