## [Editor Report · eLife assessment]

This **valuable** study showed **convincing** evidence that archerfishes can adapt their shooting behaviors to airflow perturbations. The fish also exhibits adaptive behaviors indicative of an egocentric representation of the perturbation, though direct evidence is missing. Hence, this work will be of interest to those interested in cross-species comparisons for motor learning.

---

## [Referee Report · Reviewer #1 (Public Review)]

Summary:

The authors examined whether archerfish have the capacity for motor adaptation in response to airflow perturbations. Through two experiments, they demonstrated that archerfish could adapt. Moreover, when the fish flipped its body position with the perturbation remaining constant, it did not instantaneously counteract the error. Instead, the archerfish initially persisted in correcting for the original perturbation before eventually adapting, consistent with the notion that the archerfish's internal model has been adapted in egocentric coordinates.

Evaluation:

This important study demonstrates the remarkable capacity for motor adaptation in archer fish. I found the results of both experiments to be convincing, given the observable learning curve and the clear aftereffect. Nonetheless, within the current set of experiments, no quantitative is provided to demonstrate that the archer fish is sensitive to the relative change in body position, making it unclear whether motor adaptation in archer fish indeed generalizes in egocentric coordinates.

The authors have cited a previous study to claim that archer fish are sensitive to their relative position in the water tank. However, given the absence of clear visual referents on the screen (e.g., squares with different colors in the corners) and/or some behavioral indication that the fish are sensitive to their relative change in body position, I remain sceptical of the claim that archer fish indeed generalize in egocentric rather than allocentric coordinates. The current results do not rule out the idea that archerfish are ostensibly unaware of changes in body position, they continue with previously successful actions, masquerading as egocentric generalization.

---

## [Referee Report · Reviewer #2 (Public Review)]

Summary:

The work of Volotsky et al presented here shows that adult archerfish are able to adjust their shooting in response to their own visual feedback, taking consistent alterations of their shot, here by an air flow, into account. The evidence provided points to an internal mechanism of shooting adaptation that is independent of external cues, such as wind. The authors provide evidence for this by forcing the fish to shoot from 2 different orientations to the external alteration of their shots (the airflow). This paper thus provides behavioral evidence of an internal correction mechanism, that underlies adaptive motor control of this behavior. It does not provide direct evidence of refractory index-associated shoot adjustance.

Strengths:

The authors have used a high number of trials and strong statistical analysis to analyze their behavioral data. They used an elegant experimental design in which they force the fish to shoot from directions chosen by the authors, which elegantly reduced shooting variability.

Weaknesses:

A large portion of fish did not make it to the final test (as is often the case in behavioral studies) which raises the question whether all individuals are able to solve the task.

---

## [Author Response]

The following is the authors’ response to the current reviews.

At this stage the referees had only minor comments. Referee #1 asked whether archerfish indeed generalize in egocentric rather than allocentric coordinates. It might be that the current results do not rule out the idea that archerfish are unaware of changes in body position, they continue with previously successful actions, that seems as egocentric generalization. We agree with referee #1 and updated lines 255-260 in the results and added lines 329-336 in the discussion text that mentions this possibility. Referee #2 mentioned that a portion of fish did not make it to the final test which raises the question whether all individuals are able to solve the task. We agree with referee #2 and added paragraph at the discussion section to mention this point (lines 384-388). We also added the salinity of the water in the water tanks (line 98) as per suggestion of the Referee #2. Referee #2 suggested using a different term than “washout” in the behavioral experiments. Since the term “washout” is standard in the field, we keep the term in the text.

The following is the authors’ response to the original reviews.

**eLife assessment**
This useful study explores how archerfish adapt their shooting behavior to environmental changes, particularly airflow perturbations. It will be of interest to experts interested in mechanisms for motor learning. While the evidence for an internal model for adaptation is solid, evidence for adaptation to light refraction, as initially hypothesized, is inconclusive. As such, the evidence supporting an egocentric representation might be caused by alternative mechanisms to airflow perturbations.
**Public Reviews:**

**Reviewer #1 (Public Review):**
Summary:The authors examined whether archerfish have the capacity for motor adaptation in response to airflow perturbations. Through two experiments, they demonstrated that archerfish could adapt. Moreover, when the fish flipped its body position with the perturbation remaining constant, it did not instantaneously counteract the error. Instead, the archerfish initially persisted in correcting for the original perturbation before eventually adapting, consistent with the notion that the archerfish's internal model has been adapted in egocentric coordinates.Evaluation:The results of both experiments were convincing, given the observable learning curve and the clear aftereffect. The ability of these fish to correct their errors is also remarkable. Nonetheless, certain aspects of the experiment's motivation and conclusions temper my enthusiasm.(1) The authors motivated their experiments with two hypotheses, asking whether archerfish can adapt to light refractions using an innate look-up table as opposed to possessing a capacity to adapt. However, the present experiments are not designed to arbitrate between these ideas. That is, the current experiments do not rule out the look-up table hypothesis, which predicts, for example, that motor adaptation may not generalize to de novo situations with arbitrary actionoutcome associations. Such look-up table operations may also show set-size effects, whereas other mechanisms might not. Whether their capacity to adapt is innate or learned was also not directly tested, as noted by the authors in the discussion. Could the authors clarify how they see their results positioned in light of the two hypotheses noted in the Introduction?

We agree with the referee that look up tables only confuse the issue. The question we tested is whether or not the fish uses adaptation mechanisms to correct its shooting. We have now changed the introduction both to eliminate the entire question of look up tables and also to clarify that both innate mechanisms and learning mechanisms can contribute to fish shooting, and that our research focuses on the question of whether the fish can adapt to a perturbation in its shooting caused by a change in its physical environment.

(2) The authors claim that archerfish use egocentric coordinates rather than allocentric coordinates. However, the current experiments do not make clear whether the archerfish are "aware" that their position was flipped (as the authors noted, no visual cues were provided). As such, for example, if the fish were "unaware" of the switch, can the authors still assert that generalization occurs in egocentric coordinates? Or simply that, when archerfish are ostensibly unaware of changes in body position, they continue with previously successful actions.

The fish has access to the body position switch: there are clues in a water tank that can help the fish orient inside the water tank. Additionally, there are no clues to the presence or direction of the air flow above the water tank. Moreover, previous experience has shown that the fish is sensitive to the visual cues and uses them to achieve consistent orientation within the tank when possible. These points have been added to the main text [lines 143-144, 254-257]

(3) The experiments offer an opportunity to examine whether archerfish demonstrate any savings from one session to another. Savings are often attributed to a faster look-up table operation. As such, if archerfish do not exhibit savings, it might indicate a scenario where they do not possess a refined look-up table and must rely on implicit mechanisms to relearn each time.

This is an important question. Indeed, we looked for the ‘saving’ effect in the data, but its noisy nature prevented us from drawing a concrete conclusion. We now mention this in lines 247-249.

We have also eliminated the discussion of look up tables from the article.

(4) The authors suggest that motor adaptation in response to wind may hint at mechanisms used to adapt to light refraction. However, how strong of a parallel can one draw between adapting to wind versus adapting to light refraction? This seems important given the claims in this paper regarding shared mechanisms between these processes. As a thought experiment, what would the authors predict if they provided a perturbation more akin to light refraction (e.g., a film that distorts light in a new direction, rather than airflow)?

This is an important point. Indeed, our project started by looking for options to distort the refraction index or distort the light in a new direction. However, given the available ways of distorting the light to a new direction, it is hard to achieve that on the technical level. Initially, we tried using prism goggles, however the archerfish found it hard to shoot with the heavy load on the head. We have also explored oil on the water surface. However, given the available oils and the width of the film above water, it is hard to achieve considerable perturbation.

Fish response to the perturbation matches the response to what would be expected for a change in light refraction. Light refraction perturbation does not change with the change in fish body position relative to the target. However, in response to (and in agreement with) the referees, we have generalized the context in which we see our results and discuss the results in terms of adaptation of the fish shooting behavior to changes in physical factors including light refraction, wind, fatigue, and others.

(5) The number of fish excluded was greater than those included. This raises the question as to whether these fish are merely elite specimens or representative of the species in general.

The filtering of the fish was in the training stage. The requirements were quite strict: the fish had to produce enough shots each day in the experimental setup. Very few fish succeeded. But all fish that got to the stage of perturbation exhibited the adaptation effect. We do not see a reason to think that the motivation to shoot will have a strong interaction with the shooting adaptation mechanisms.

**Reviewer #2 (Public Review):**
Summary:The work of Volotsky et al presented here shows that adult archerfish are able to adjust their shooting in response to their own visual feedback, taking consistent alterations of their shot, here by an air flow, into account. The evidence provided points to an internal mechanism of shooting adaptation that is independent of external cues, such as wind. The authors provide evidence for this by forcing the fish to shoot from 2 different orientations to the external alteration of their shots (the airflow). This paper thus provides behavioral evidence of an internal correction mechanism, that underlies adaptive motor control of this behavior. It does not provide direct evidence of refractory index-associated shoot adjustance.Strengths:The authors have used a high number of trials and strong statistical analysis to analyze their behavioral data.Weaknesses:While the introduction, the title, and the discussion are associated with the refraction index, the latter was not altered, and neither was the position of the target. The "shot" was altered, this is a simple motor adaptation task and not a question related to the refractory index. The title, abstract, and the introduction are thus misleading. The authors appear to deduce from their data that the wind is not taken into account and thus conclude that the fish perceive a different refractory index. This might be based on the assumption that fish always hit their target, which is not the case. The airflow does not alter the position of the target, thus the airflow does not alter the refractive index. The fish likely does not perceive the airflow, thus alteration of its shooting abilities is likely assumed to be an "internal problem" of shooting. I am sorry but I am not able to understand the conclusion they draw from their data.

This is an important point. Indeed, our project started by looking for options to distort the refraction index or distort the light in a new direction. However, given the available ways of distorting the light to a new direction, it is hard to achieve that on the technical level. Initially, we tried using prism goggles, however the archerfish found it hard to shoot with the heavy load on the head. We have also explored oil on the water surface. However, given the available oils and the width of the film above water, it is hard to achieve considerable perturbation.

Fish response to the perturbation matches the response to what would be expected for a change in light refraction. Light refraction perturbation does not change with the change in fish body position relative to the target. However, in response to (and in agreement with) the referees, we have generalized the context in which we see our results and discuss the results in terms of adaptation of the fish shooting behavior to changes in physical factors including light refraction, wind, fatigue, and others.

**Reviewer #2 (Recommendations For The Authors):**
I have had a hard time trying to understand how the authors concluded that the RI is important here as it is not altered. Thus I did not understand the conclusions drawn from this paper. The experiments are well described, but the conclusions are not to me. Maybe schematics would help to clarify. I am from outside the field and represent a naïve reader with an average intellect. The authors need to do a better job of explaining their results if they want others to understand their conclusions.

See response to the public comments.

Minor comments:Line 9: omit the "an".

Done.

Line 11: this sentence would fit way better if it followed the next one.Done.Line 15: and all the rest of the paper: washout is a strange term and for me associated with pharmacological manipulations - might only be me. I suggest using recovery instead throughout the manuscript.

The term ‘washout’ is often used in the field of motor adaptation to describe the return to original condition. For example:

Kluzik J, Diedrichsen J, Shadmehr R, Bastian AJ (2008) Reach adaptation: what determines whether we learn an internal model of the tool or adapt the model of our arm? J Neurophysiol 100:1455-64. doi:10.1152/jn.90334.2008

Donchin O, Rabe K, Diedrichsen J, Lally N, Schoch B, Gizewski ER, Timmann D (2012) Cerebellar regions involved in adaptation to force field and visuomotor perturbation. J Neurophysiol 107:134-47

Line 19: the fish does not expect the flow, it expects that it shoots too short- no?

Done.

Line 35: fix the citation - in your reference manager.

Done.

Line 52: provide some examples of the mechanisms you think of or papers of it for naive readers. Otherwise, this sentence is not helpful for the reader.

Done.

Line 183: it's unclear which parameter you mean. Rephrase.

Done.

Line 197: should read to test "the" - same sentence: you repeat yourself- rephrase the sentence.

Done.

Figure 4: it was unclear to me why the figure was differentiating between fishes until I read the legend. Why not include direct information in the figure? A schematic maybe? Legend: you have a double "that" in C.

We added the title for each column with the information about the direction of air.

Figures: in all figures, perturbation is wrongly spelled! Change the term washout to recovery.

Done. We kept the term ‘washout’